# Fabrication of Vertical MEMS Actuator with Hollow Square Electrode for SPR Sensing Applications

**DOI:** 10.3390/s22239490

**Published:** 2022-12-05

**Authors:** Kihyun Kim, Yeonsu Lee, Ignacio Llamas-Garro, Jung-Mu Kim

**Affiliations:** 1School of Electronic and Information Engineering, Jeonbuk National University, Jeonju 54896, Republic of Korea; 2Division of Electronic Engineering, Jeonbuk National University, Jeonju 54896, Republic of Korea; 3Samsung Electro-Mechanics, Sejong 30067, Republic of Korea; 4Centre Tecnològic de Telecomunicacions de Catalunya, CTTC/CERCA, 08860 Castelldefels, Spain

**Keywords:** vertical MEMS actuator, hollow square electrode, SiOG, Otto configuration, surface plasmon resonance

## Abstract

In this study, an electrostatically driven vertical MEMS actuator was designed using a hollow square electrode. To attain vertical actuation, a hollow square-shaped electrode was designed on the glass substrate. The silicon proof mass, containing a step, was utilized to realize analogue actuation without pull-in. The vertical MEMS actuator was fabricated using the SiOG (Silicon on Glass) process and the total actuator size was 8.3 mm × 8.3 mm. The fabricated proof mass was freestanding due to eight serpentine springs with 30 μm width. The vertical movement of the MEMS actuator was successfully controlled electrostatically. The measured vertical movement was 5.6 µm for a voltage of 40 V, applied between the top silicon structure and the hollow square electrode. The results shown here confirm that the proposed MEMS actuator was able to control the vertical displacement using an applied voltage.

## 1. Introduction

MEMS actuators are useful devices that convert the input energy into physical motion in the mechanical domain. They have attracted much attention in many areas including medical [1,2], automotive [3,4], electronics communication [5], environment monitoring [6] fields and many more with a potential market of billions of dollars, since MEMS actuators perform accurate physical functions. The required features for MEMS actuators are high compatiblity with conventional IC technology and high reliability. Single–crystalline silicon (SCS) is an attractive candidate for fabricating MEMS structures due to its advantages such as lack of residual stress, low mechanical fatigue, and reduced thermal effect compared with other materials [7]. As the demand for MEMS actuators with large deflection has increased, the fabrication method, e.g., the silicon-on-glass (SiOG) process, has been developed [8]. The SiOG substrate can be simply made by using the anodic bonding of silicon and glass wafers. The popular mechanisms that drive actuation are categorized into four types: electrostatic, electromagnetic, piezoelectric, and thermal expansion [9,10,11,12]. The MEMS actuators fabricated using the SiOG process are operated by electrostatic actuation, which is based on the attraction of two oppositely charged plates on the silicon and glass wafers. Compared to other actuation mechanisms, the electrostatic method has the advantage of not requiring a special condition for fabrication. The silicon actuator is only made to be placed with an air gap above the electrode on the glass substrate and it has been widely used in the application of optical MEMS such as micro-mirrors [13]. In general, the light is incident on the reflective metal on the top silicon substrate, and the electrode for vertical actuation is formed on the bottom glass substrate, as shown in Figure 1a. Thus, the air gap of the conventional vertical MEMS actuator only plays the role of providing vertical movement, therefore the incident light is not able to penetrate and reach the bottom glass substrate using the conventional design, containing a bottom electrode for electrostatic actuation. Another application where silicon actuators are most widely used is in surface plasmon resonance (SPR) sensors. SPR sensors have recently attracted much attention as a powerful tool due to their advantages of label-free and real-time detection [14,15,16,17,18]. The SPR sensor is widely used in many fields such as chemical sensing, foodborne marker screening, environmental monitoring, and medical diagnostics [19]. Surface plasmons are basically the charge density oscillations that occur by coupling the evanescent wave produced by total internal reflection with the collective oscillation of electrons at a metallic surface. The surface plasmons are very sensitive to the environment refractive index change induced by chemical species and sensing analytes. There is a promising actuator configuration used for SPR sensors, called Otto configuration, where the air gap distance should be approximately micron size (around two times the wavelength). The air gap in Otto configuration provides the actuation space and defines the sensing channel for gas or bio-sensing material. The incident light should come from the bottom glass side after attachment to a prism for specific SPR conditions, as shown in Figure 1b. The hollow square electrode on the bottom glass substrate is used for both vertical actuation and optical window, allowing light penetration due to the transparent bottom substrate. In Otto configuration, when the air gap between silicon and glass wafers is adjusted, the resonance wavelength of SPR can be linearly controlled by increasing the air gap [20]. This makes it possible to develop a sensor that can detect various target gas and molecules with one reconfigurable SPR sensor chip.

In this study, we report a vertically actuated MEMS device with a hollow square electrode. The actuator is fabricated using the SiOG process, which is compatible with the conventional semiconductor fabrication process. Vertical displacement characteristics of the fabricated MEMS actuator were measured with respect to the applied voltage. In addition, the measured characteristic was analyzed in detail.

## 2. Design and Fabrication

### 2.1. Design of the Vertical MEMS Actuator

Figure 2 illustrates a schematic view of the proposed vertical MEMS actuator based on the Otto configuration. The actuator consists of a low-resistivity (LR) silicon substrate (Figure 2b) which includes a proof mass, eight serpentine springs, two anchors, and a ground electrode, as well as a glass substrate (Figure 2c), which includes the hollow square electrode. In the LR silicon substrate, the proof mass is suspended by eight serpentine springs connecting two anchors (Figure 2a). To support the actuator plate, various types of springs such as crab-leg spring [21], clamped-clamped spring [22], folded-beam spring [23], and serpentine spring [24] have been designed and used in MEMS actuators. Among them, we used the serpentine spring for the design of the vertical MEMS actuator, because it can provide larger vertical deflection due to a low vertical direction spring constant per the unit area [25]. In case of the glass substrate, the hollow square electrode has been proposed for SPR sensor application (see Figure 2b). For the SPR sensor application, the light generated by the laser must be incident on the glass substrate and then arrive on the silicon actuator. The electrode on the glass substrate should be able to pass light with little loss and induce actuator movement by applying a voltage. The hollow square electrode is a suitable structure that satisfies all of these conditions. To induce electrostatic actuation, the hollow square electrode comes into contact with the ground electrode on the LR silicon substrate. For the vertical actuation, while the drive voltage is applied to the anchors, which connect with the proof mass through the serpentine spring, the hollow square electrode is connected to the ground through the silicon ground electrode using anodic bonding. The proof mass is actuated in the vertical direction by the voltage difference between the anchor and the ground electrode, resulting in a reduced air gap.

The design parameters of the vertical MEMS actuator are presented in Figure 3. The details of the design parameters are summarized in Table 1. The proof mass is a square plate with a side length of 4 mm. Most notably, the proof mass is designed in the form of a step where thickness is not constant. The maximum and minimum thickness of the proof mass are 52 μm and 40 μm, respectively. The step-shaped proof mass can be analogously moved without pull-in because its structure can separate the light reflection area and vertical actuation area. The resonance frequency is one of the most important parameters to consider when designing the MEMS actuator. If the MEMS actuator has a very low resonance frequency, the device is easily damaged by any external mechanical shock. On the other hand, when the resonance frequency is high, the voltage required for actuation becomes large. A resonance frequency of 1.3 kHz was chosen after considering the feasibility of fabrication and the specifications of measurement equipment.

This resonance frequency can prevent actuator damage and demonstrate the actuator’s displacement with our laboratory equipment. The resonance frequency of the proof mass is expressed as follows:(1)$$f=\frac{1}{2\pi }\sqrt{\frac{k}{m}}$$
where *m* is the weight of the proof mass and *k* is the spring constant. The proof mass weight is 1.7 × 10^−6^ kg, which is calculated using the silicon density (2.329 g/cm^3^) and the proof mass volume. Using Equation (1) and the resonance frequency of 1.3 kHz, we found that the actuator should have a spring constant of 113 N/m. The serpentine spring dimension was designed to have the calculated spring constant value. When the serpentine spring has a symmetric structure, i.e., the number of connector beams are even, only the spring constant in the z-axis direction is considered for spring design due to the guided-end boundary condition. The spring constant in the z-axis direction is calculated as follows [26]:(2)$$k_{z}=\frac{48S_{ea}S_{eb}S_{ga}S_{gb}}{\left\{\begin{array}{c} S_{eb}S_{ga}a^{2}\left(S_{gb}a+S_{ea}b\right)n^{3}-3S_{ea}S_{eb}S_{ga}a^{2}bn^{2}+\\ S_{ea}b\left(2S_{eb}S_{ga}a^{2}+3S_{eb}S_{gb}ab+S_{ga}S_{gb}b^{2}\right)n-S_{ea}S_{ga}S_{gb}b^{3} \end{array}\right\}}$$
(3)$$S_{ea}\equiv EI_{x,a},\;S_{eb}\equiv EI_{x,b},\;S_{ga}\equiv GJ_{a},\;S_{gb}\equiv GJ_{b}\;$$
(4)$$I_{x,a}=I_{x,b}=\frac{t^{3}w_{s}}{12},\;G=\frac{E}{2\left(1+\nu \right)},\;J_{a}=J_{b}=\frac{1}{3}t^{3}w_{s}\left(1-\frac{192t}{\pi^{5}w_{s}}\sum_{i=1}^{\infty }\frac{1}{\pi^{5}}tanh\left(\frac{i\pi w_{s}}{2t}\right)\right)$$
where *S_e_* is Elastic modulus, *S_g_* is torsional constant, *n* is the number of connector beams, *E* is Young’s modulus of silicon (169 GPa), *I_x_* is the moment of inertia in the x-direction, *G* is shear stress, *J* is the torsional cross-section constant of the beam, *t* is beam thickness, and ν is Poisson’s ratio (0.23). Through calculation, the spring width, connector beam length, and length of the serpentine spring span beam were set to 30 μm, 80 μm, and 1.71 mm, respectively (Table 1).

In order to realize the proposed MEMS actuator, the Otto configuration is made by using the anodic bonding process of the LR silicon wafer and glass wafer. In the anodic bonding process, the LR silicon wafer is bonded to the glass wafer by applying an external high voltage under proper temperature. The electrostatic force generated by applying an external high voltage is the prerequisite for the bonding reaction to occur. However, this high voltage can cause the dielectric breakdown when the air gap is too narrow. In 2010, Tirumala et al. reported the effect of electrode spacing on the dielectric breakdown [27]. Considering our equipment specifications, we concluded that we should conduct the anodic bonding process at 300 V. According to Tirumala’s report, the electrode spacing is less than 5 μm when the dielectric breakdown is 300 V. The MEMS actuator displacement is limited by the physical properties of silicon and the pull-in voltage [28]. For the SPR sensor application, it is important that the air gap should be set to a suitable value by considering the wavelength used in the SPR setup. The SPR with the gold film can induce an efficient resonance phenomenon when light ranges from visible to infrared. Therefore, in the SPR sensor application, the air gap of the MEMS actuator is designed to be greater than at least 2 μm to induce the resonance condition according to the air gap at 980 nm wavelength. The distances between actuation electrodes on the silicon substrate and the gold electrode on the glass substrate are set to 18 μm after considering pull-in voltage. Considering the conditions mentioned above, the air gap distance was set to 6 μm (1/3 of 18 μm). This means that the actuator can move up to 6 μm analogously in the vertical direction.

To predict the MEMS actuator vertical displacement, the spring-mass-damper system needs to be understood. The proof mass is physically suspended by the serpentine springs connecting the anchor in the MEMS actuator design, as shown in Figure 4a. When the drive voltage is applied between the proof mass and the bottom electrode, the proof mass is vertically moved by the electrostatic force generated by drive voltage. During vertical actuation, the proof mass experiences attenuation of displacement induced by resistance to air and anchors. The gravity due to mass is generated in the downward direction of the proof mass, causing an additional driving force. Therefore, the electrostatic displacement of the proposed MEMS actuator is expressed as follows:(5)$$mx^{\prime \prime }+bx^{\prime }+kx=F_{E}+F_{g}$$
(6)$$mx^{\prime \prime }+bx^{\prime }+kx=\frac{\epsilon_{0}AV^{2}}{2{\left(g_{air}-x\right)}^{2}}+g_{air}mg$$
where *ε*_0_ is the permittivity of a vacuum, *A* is the area where electrodes are faced and where the potential difference occurs, *g* is the gravitational acceleration.

The electrostatic-driven behavior of the proposed actuator has been analyzed using numerical calculation in MATLAB Simulink (version R2013b) and FEM simulation in COMSOL Multiphysics (version 4.3). For the numerical calculation, the vertical displacement was calculated by the design parameters and Equation (6) in MATLAB Simulink. The calculated z-axis spring constant and the desired resonance frequency are used as the design parameters in this work. The calculated displacement is 4.9 μm when the drive voltage of 60 V is applied between the proof mass and electrode. In FEM simulations, it is assumed that the electrostatic force and gravity are applied to the proof mass in the +z-axis direction, so the displacement occurs in the +z-axis. According to the simulation results, the vertical displacement is 0.13 μm, 2.26 μm, 3.54 μm, and 5.1 μm for drive voltages of 0 V, 50 V, 55 V, and 60 V, respectively, as shown in Figure 4b. It can be seen that, although the potential difference is 0 V, the displacement of 0.13 μm is generated by the gravity. However, the displacement does not significantly affect the electrostatic-driven behavior because it is about 1/17 times smaller than 50 V case. Figure 4c shows the displacement comparison obtained from the FEM simulation and the numerical calculation. It turns out that the calculated result is similar to the simulated result because the difference is only 3%. In the case of the numerical calculation, the spring constant model for the serpentine spring is simplified under various assumptions, one being that non-ideal phenomena do not occur in order to minimize computational complexity. For example, only the spring constant in the z-axis direction is considered for vertical displacement prediction. On the other hand, the FEM simulation predicts the vertical displacement by considering boundary conditions similar to the real environment. The small difference between the predicted results obtained by the two methods means that the design model utilized for the serpentine spring is very reliable. Additionally, we observed the pull-in from 6 μm displacement (1/3 of 18 μm) in the simulation, as we expected.

### 2.2. Fabrication of the Vertical MEMS Actuator

The proposed vertical MEMS actuator is fabricated using the SiOG process, which is widely used in the MEMS field. A 725 μm-thick LR silicon wafer (p-type, 8–12 Ω·cm) and a 500 μm-thick glass wafer were used as starting materials. The fabrication process flow is presented in Figure 5. First, the step-shaped proof mass was formed on the LR silicon wafer. A 500 nm-thick SiO_2_ film was deposited on the front-side surface to serve as a hard mask and then a square pattern with an area of 2.9 mm × 2.9 mm was defined using conventional photolithography and positive photoresist. The SiO_2_ film was etched using an inductively coupled plasma reactive ion etching (ICP-RIE). After the end of the etching process, the remaining photoresist was removed through O_2_-plasma ashing. The second pattern was formed using a negative photoresist with a high thickness (10 μm). The silicon layer on the patterned wafer was etched to a depth of 12 μm using ICP-RIE, followed by etching the SiO_2_ hard mask using a diluted hydrofluoric acid solution. Then, in order to form the step profile, the silicon layer was additionally etched to a depth of 6 μm. The maximum depth is 18 μm and the height difference of the step is of 12 μm. In order to form the electrode of the actuator, 10 nm-thick Cr and 200 nm-thick Au layers were selectively deposited in square shapes using the shadow mask, not lift-off. In the case of the lift-off process, when the photoresist is coated on the surface of the step structure, the area where the photoresist is not coated can be observed due to high step height. The shadow mask is the best method to avoid this coating problem. Second, the hollow square electrode was defined on the glass wafer. The hollow square electrode was patterned using the negative photoresist. The 10 nm-thick Cr and 200 nm-thick Au layers were deposited using an electron-beam evaporator, followed by the lift-off process. Third, the prepared glass wafer was anodically bonded to the silicon wafers by applying a voltage of 300 V at 300 °C. The anodic bonding was successful without the dielectric breakdown. Fourth, the serpentine spring was fabricated on the silicon wafer. The silicon wafer was thinned by a grinding process until the thickness was brought down to 58 μm. Next, the serpentine spring pattern made of 500 nm-thick Al was formed using the electron-beam evaporator and the lift-off to be utilized as a hard mask. In order to make the serpentine spring, the silicon layer was etched to a depth of more than 40 μm using deep reactive ion etching (DRIE). Finally, the Al layer was entirely removed by immersion in Al etchant. The size of the fabricated MEMS actuator is 8.3 mm × 8.3 mm.

## 3. Results and Discussion

These scanning electron microscope (SEM) images of the fabricated MEMS actuator are shown in Figure 6. The serpentine spring was successfully formed between the proof mass and the anchor, as shown in Figure 6a–c. As we can see in Figure 6d, although the width of serpentine spring was designed to be 30 μm, the actual width is smaller than 30 μm and the edge of serpentine spring is severely rough. The maximum width is 29.8 μm and the minimum width is 24.4 μm, which means that 22% variation has been induced due to the footing effect and thermal isolation [29]. These effects are negligible in most general etching systems. However, it becomes more evident in the DRIE systems because it generates high-density plasma [30]. Design parameters such as proof mass width, spring width, connector beam length, span beam length, and air gap are in agreement with the designed values when these variations are not considered. However, the fabricated device shows a large difference compared to the design value in some parameters. The serpentine spring is suspended from the surface of the glass substrate with the distance between electrode (*g_elec_*) of 14 μm (4 μm longer than the design value), as shown in Figure 6e. It turns out that the proof mass thickness and spring are 78.4 μm and 70 μm, respectively, which are thicker than the designed value by 26.4 μm and 30 μm, respectively. The step height on the proof mass is 8.4 μm (Figure 6f). Therefore, the air gap (*g_air_*) between the Au electrode on the proof mass and the glass substrate is estimated to be approximately 5.6 μm. The designed parameters and the actual measured results show slight differences. In the fabricated actuator, this divergence from the designed values caused the change in the theoretically predicted electrostatic-driven behavior.

The displacement of the fabricated MEMS actuator is measured using a confocal microscope (NanoFocus, Oberhausen, Germany) as shown in Figure 7a. In order to measure the proof mass displacement, various static drive voltages were applied to the proof mass through the anchors, while the hollow square electrode was connected to the ground through the silicon ground electrode. The applied drive voltage was increased from 0 V to 40 V with an interval of 10 V. The vertical displacement was measured using an objective lens including an LED source and CCD camera. Figure 7b shows the vertical displacement according to the applying drive voltage. The measured displacement increases from 0.24 to 5.6 μm when the drive voltage increases from 10 to 40 V. The proof mass moves towards the downward direction when a positive voltage is applied. This measured displacement is larger than the theoretically predicted results (4.9 μm displacement at 60 V for numerical calculation and 5.1 μm displacement at 60 V for FEM simulation). It can be seen that the vertical displacement increases with increasingly applied drive voltage. However, the relationship between vertical displacement and drive voltage shows nonlinear behavior, indicating that the incremental displacement for each drive voltage change is 0.24 μm for 10–20 V, 0.75 μm for 20–30 V, and 4.37 μm for 30–40 V, as shown in Figure 7c. The increasing difference of measured displacement becomes higher as the drive voltage increases. The proof mass suddenly pulls-in when 40 V is applied. In COMSOL simulation, the pull-in phenomenon occurs at 69 V, which is 1.7 times higher than the measured data, even when the measured dimension values are reflected. The inconsistency between simulation and measurement results is attributed to the non-ideal phenomenon as well as the changes in design parameters. The serpentine spring seems to be formed with a thickness of 70 μm and a smooth surface in Figure 6e,f, but the thickness variation is large and the surface is rough due to the footing effect and thermal isolation [29]. In addition, the thickness and width of the spring decreased, moving further from the anchor, and the top-view shape changed from rectangular to a reverse trapezoid due to side etch during the DRIE process. Thus, we believe that these non-ideal phenomena caused the spring constant to be much smaller than expected, resulting in pull-in voltage reduction. However, we could see the analogous actuation before 30 V stably before the pull-in. When the drive voltage is kept constant, the displacement hardly changes even if the same drive voltage is applied several times before applying the pull-in voltage of 40 V. This means that the fabricated actuator can operate stably, as shown in Figure 7b.

## 4. Conclusions

We have designed, fabricated, and measured a proposed vertical MEMS actuator containing a hollow square electrode for potential SPR sensing applications. The vertical MEMS actuator consists of a proof mass, eight serpentine springs, two anchors, ground electrode, and a hollow square electrode. The vertical MEMS actuator was fabricated using the SiOG process. The hollow square electrode fulfils the role of providing vertical actuation and a transparent optical window on the glass substrate. The fabricated actuator shows analog vertical displacement before pull-in. The vertical displacement characteristic was theoretically analyzed by comparing simulation results with numerical calculation results. The proposed vertical MEMS actuator has great potential in the field of optical sensing systems with small gaps, such as a gap-tunable SPR sensor using the Otto configuration.

## Figures and Tables

**Figure 1 sensors-22-09490-f001:**
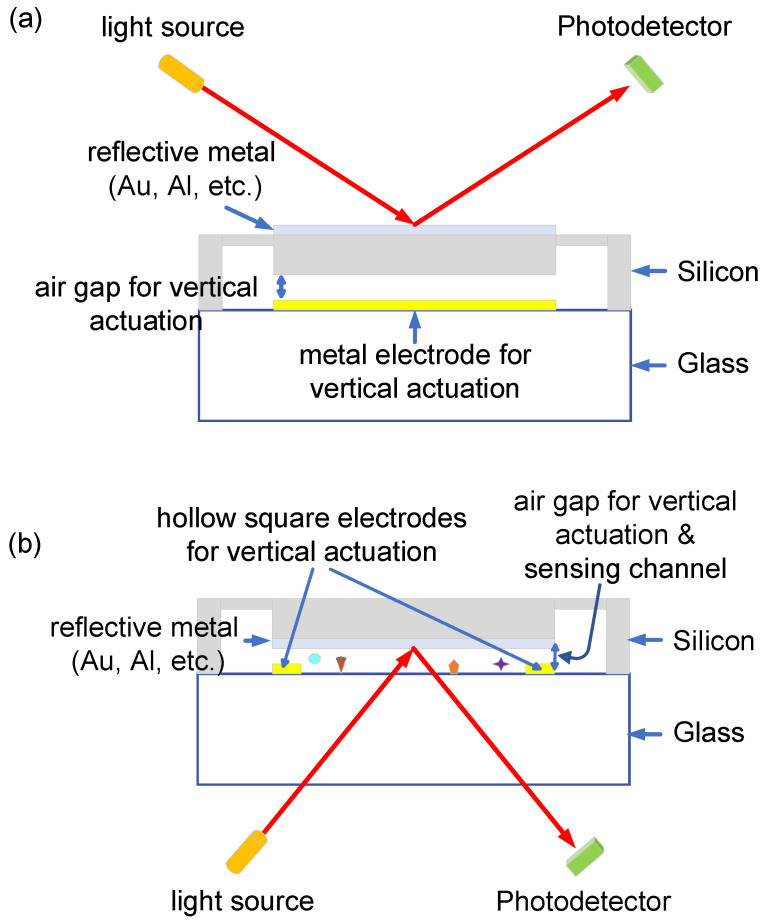
Optical MEMS device structure. (**a**) general structure and (**b**) structure with hollow square metal electrode for Otto configuration.

**Figure 2 sensors-22-09490-f002:**
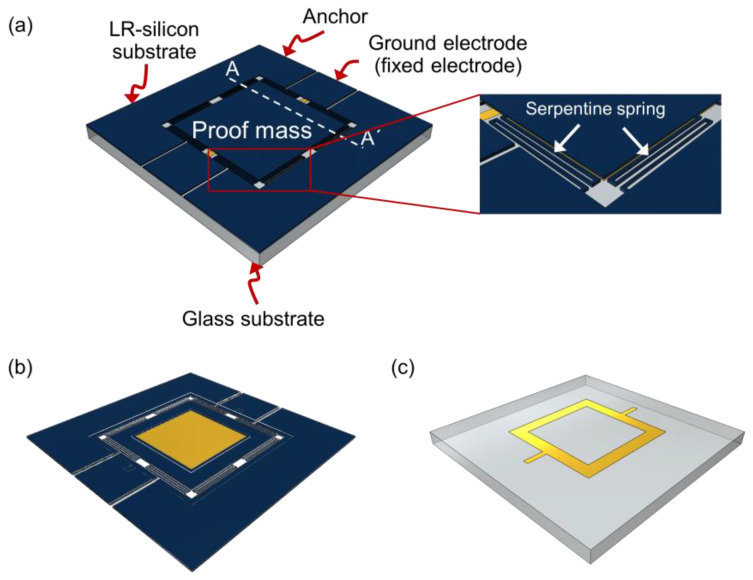
(**a**) Schematic diagram of vertical MEMS actuator based on Otto configuration, (**b**) silicon substrate viewed from the side for anodic bonding, (**c**) glass substrate viewed from the side for anodic bonding.

**Figure 3 sensors-22-09490-f003:**
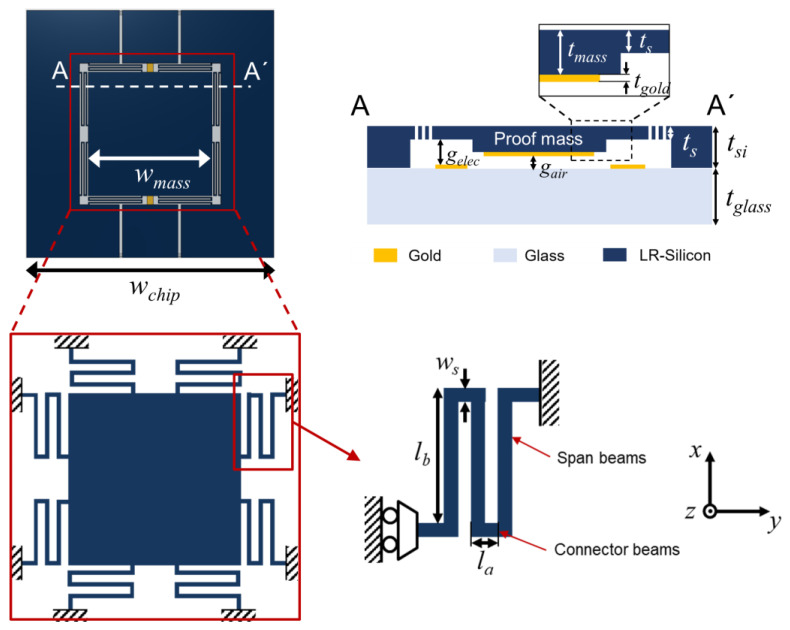
Vertical MEMS actuator design parameters. All parameters are defined in the top-view and cross-view schematics. The cross-view is drawn based on the cutline of AA′.

**Figure 4 sensors-22-09490-f004:**
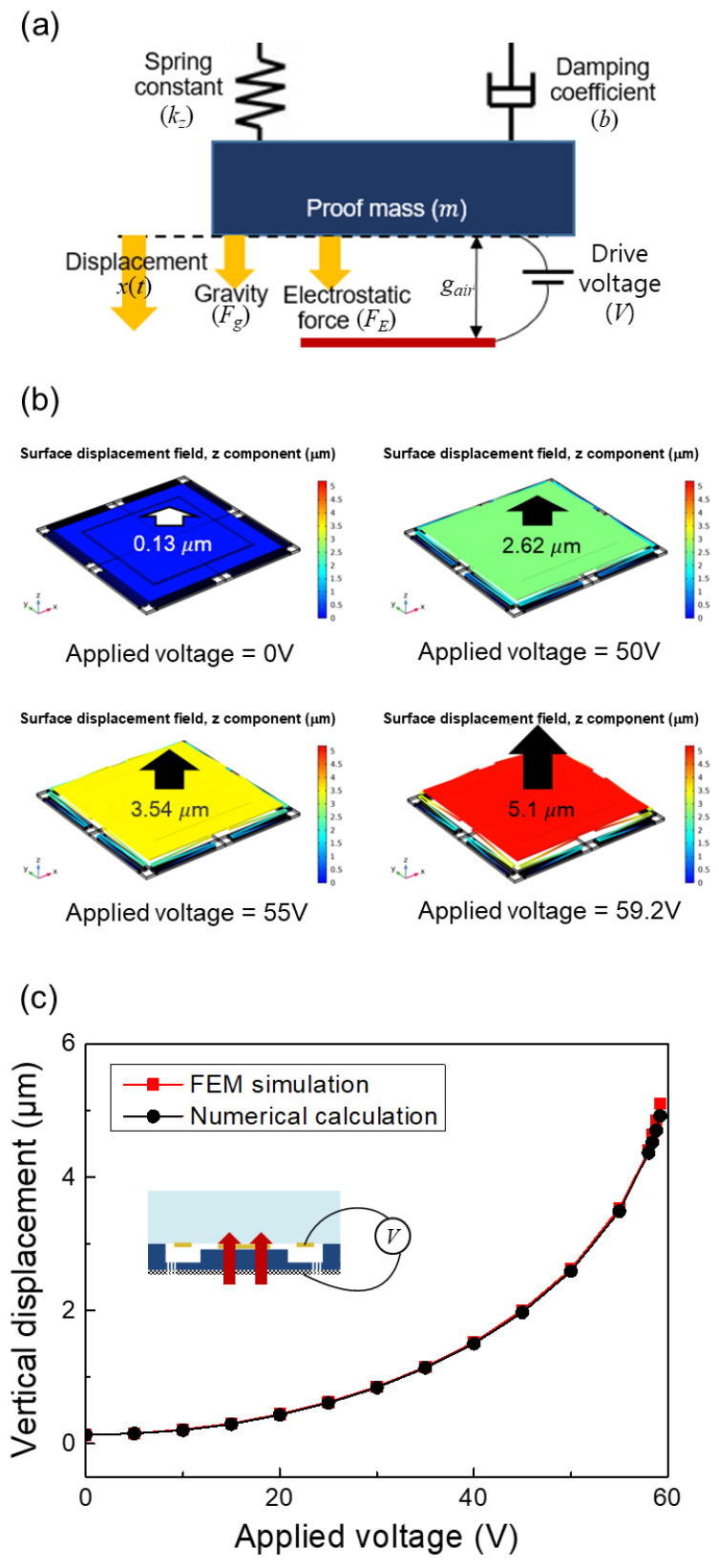
(**a**) Vertical MEMS actuator schematic expressed by spring-mass-damper system, (**b**) simulation results of electrostatic-driven behavior, (**c**) comparison of displacement obtained by the FEM simulation and the numerical calculation.

**Figure 5 sensors-22-09490-f005:**
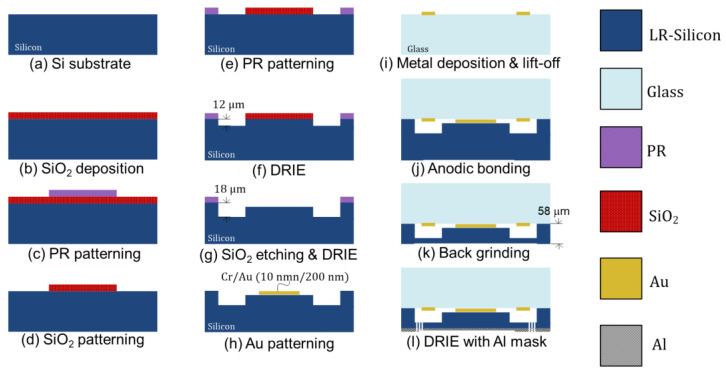
Fabrication process flow for vertical MEMS actuator.

**Figure 6 sensors-22-09490-f006:**
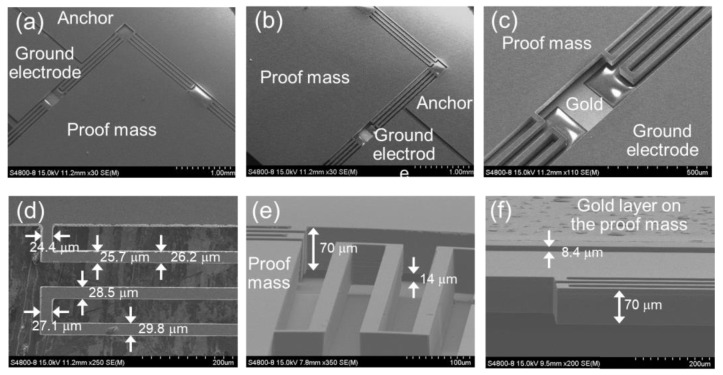
SEM images of the fabricated vertical MEMS actuator. (**a**,**b**) serpentine spring and anchor part, (**c**,**d**) close-up view of the serpentine spring part, (**e**) tilt view of the serpentine spring part when the glass substrate is positioned on the bottom side, (**f**) tilt view of the serpentine spring part when the gold layer on the proof mass is positioned on the top side.

**Figure 7 sensors-22-09490-f007:**
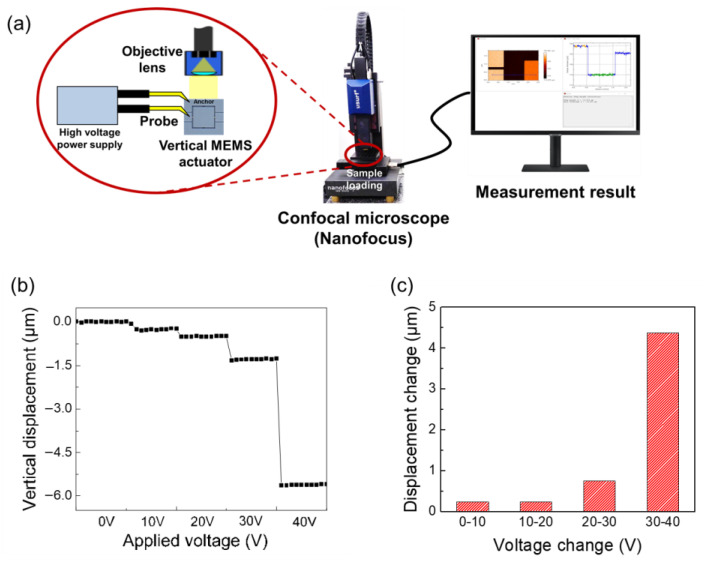
(**a**) Measurement set up schematic, (**b**) measured vertical displacement according to the applied voltage, (**c**) displacement change with voltage change of 10 V intervals.

**Table 1 sensors-22-09490-t001:** Design parameter values for the vertical MEMS actuator.

Part	Symbol	Description	Value
Proof mass	*w_mass_*	width of proof mass	4 mm
	*t_mass_*	thickness of proof mass	52 μm
	*t_gold_*	thickness of gold film	200 nm
Serpentine spring	*w_s_*	width of serpentine spring	30 μm
	*l_a_*	length of connector beam	80 μm
	*l_b_*	length of span beam	1.71 mm
	*t_s_*	thickness of serpentine spring	40 μm
Chip	*w_chip_*	width of chip	8.3 mm
	*t_si_*	thickness of silicon substrate	58 μm
	*t_glass_*	thickness of glass substrate	500 μm
	*g_air_*	distance of air gap	6 μm
	*g_elec_*	distance between electrodes	18 μm

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
