# Peer review of "Fabrication of Vertical MEMS Actuator with Hollow Square Electrode for SPR Sensing Applications"

_sensors, 2022, doi:10.3390/s22239490_

Round 1

Reviewer 1 Report

The authors report a works where a MEMS actuator was developed. In general the manuscript is well written and with a good structure. The introduction is detailed enough to frame the current challenges, and what the manuscript addresses. However, the paper may lack some comparisons with similar works, and in what this actuator is different from existing ones. The bibliography can also have more references. 

The article merits to published, after addressing some minor issues. 

1) The authors did a very good work on the methodology description. Nevertheless, it is not clear how the serpentine spring dimension was designed to have a 122 spring constant of 113 N/m ? 

2) Additionally, It is not clear why COMSOL simulations and MATLAB calculations where thoroughly compared, but then the experimental results and the simulations are just briefly mentioned, but not discussed discussed in great extent.  

Typo:

 line 130,  otto should be capitalized. 

Author Response

We thank the reviewers for their valuable comments, which have helped us to further improve the clarity of our manuscript. The revised manuscript takes into account the suggestions from the reviewers. Below, we provide a point-by-point response for your convenience.

Black: Reviewer comments
Blue: Our response
Red: Revisions made to the manuscript

Reviewer 2 Report

The Authors present a MEMS-based surface plasmon resonance sensing. The data presented here is sufficiently novel to be published. However, there are some mandatory changes to improve the manuscript. 

1) equation 1 is too simple to be used as a model. Better models are available in the literature and should be used.

2) Data presented in figure 7 seems to be of one device. The performance of the device must be studied with more rigor by testing multiple samples and reporting the standard deviation. The influence of environmental factors like humidity and temperature is not considered. What would be their effect? If not experimental, it can be analyzed theoretically.

Author Response

(The authors gave the same response as above.)

Reviewer 3 Report

This paper designed, fabricated and measured a proposed vertical MEMS actuator containing a hollow square electrode for potential SPR sensing applications. In this study, four processes of theoretical calculation, simulation, process manufacturing and test verification are carried out for the designed structure, and the reasons for the difference between the theoretical results and the test results are analyzed. The resulting actuator can apply 40V electrostatic voltage between the top silicon structure and the hollow square electrode, and the measured vertical displacement is 5.6μm. The proposed work has a potential in the field of optical sensing systems.

Suggestions:

1. The contents of lines 168-184 are exactly the same as those of lines 190-206.

2. It is suggested to establish a physical model according to the relationship diagram between applied voltage and vertical displacement in FIG. 4 and FIG. 7;

3. Further optimization measures for the pull-in effect at 40V by inferring that the spring constant is smaller than expected should be further analyzed.

Author Response

(The authors gave the same response as above.)

Reviewer 4 Report

In the paper "Fabrication of vertical MEMS actuator with hollow square electrode for SPR sensing applications" the authors did a very good job. The work carefully respects all the written and unwritten rules of a quality scientific paper.

 A first critical remark is related to the size of some essential figures that I think have been exaggeratedly reduced for paging reasons. Unfortunately, this aspect of graphic nature depreciates the appearance of the otherwise neatly elaborated work.

 A second observation that would bring a benefit to the way of presentation is the inclusion in the introduction of information in which to highlight the novelties and elements of originality of this approach. This is important given the abundance of articles on this area.

 I also consider a broadening of the references with the close theme in order to provide readers with the necessary elements for an appreciation of the elements of originality in your work.

 Some aspects of a tutorial nature would be welcome to help the researchers in the field to be able to reproduce the results published by you. The field approached is of great interest and I consider this work to be a beneficial contribution.

Author Response

(The authors gave the same response as above.)

Round 2

Reviewer 2 Report

The manuscript can be accepted

Reviewer 4 Report

The authors of the paper "Fabrication of vertical MEMS actuator with hollow square electrode for SPR sensing applications" in the revised form answered the previous observations completely. Also, the paper is in this form much improved. A small observation is related to the text around  Figure 6 being generated by a careless drafting. This small shortcoming is easy to solve during editing. The work can be accepted in this form.